# Barriers to Cardiac Rehabilitation among Patients Diagnosed with Cardiovascular Diseases—A Scoping Review

**DOI:** 10.3390/ijerph21030339

**Published:** 2024-03-13

**Authors:** Pupalan Iyngkaran, Pavithra Yapa Appuhamilage, Gayani Patabandige, Prasadi Saubhagya Sarathchandra Peru Kandage, Wania Usmani, Fahad Hanna

**Affiliations:** 1Torrens University Australia, Melbourne, VIC 3000, Australia; 2University of Notre Dame, Melbourne, VIC 3000, Australia; 3Program of Public Health, Department of Health and Education, Torrens University Australia, Melbourne, VIC 3000, Australia; yapa.pavithra@gmail.com (P.Y.A.); wimanshaprasad@gmail.com (G.P.); perusarathchandra@gmail.com (P.S.S.P.K.);

**Keywords:** barriers, cardiac rehabilitation, cardiovascular diseases, prevention, scoping review

## Abstract

Background: Cardiovascular diseases (CVDs) are a rising global burden. Preventative strategies such as cardiac rehabilitation (CR) have shown a marked reduction in disease burden. Despite this, CR is underutilized worldwide. This study aims to identify the barriers to CR among patients diagnosed with CVD. Methods: A scoping review of the literature was conducted following the Joanna Briggs Institute (JBI) guidelines. Four major databases, including CINAHL, PubMed, EBSCOhost, and Scopus, were used to obtain studies published between 2010 and 2023. Search terms such as “Cardiac rehab*”, “Barrier*”, “Cardiovascular”, “Disease”, and “diagnosis*” were utilized in order to obtain subject-specific studies relevant to the research question. Results: From the initial 2098 studies, only 14 were included in the final analysis, consisting of both qualitative and quantitative designs. The thematic analysis included “healthcare system-related factors”, “Socioeconomic factors”, and “individual characteristics”. Healthcare system-related factors were mostly related to the poor availability of CR programs, lack of proper referral strategies, inadequate knowledge of CR provider and inter-provider communication issues, and lack of alternative methods of CR delivery. The socioeconomic barriers were lack of education, longer distance to CR facilities, high cost of care, unemployment, and poor income status. The identified individual characteristics were female gender, older age, and comorbidities. Conclusions: Lack of resources, poor access, educational attainment, and high cost of care were some of the barriers to CR, particularly in low- and middle-income countries (LMICs). Health policymakers and healthcare providers should implement strategies incorporating the issues identified in this scoping review. Systematic reviews may be required to confirm these findings.

## 1. Introduction

Cardiovascular diseases (CVDs) are a range of disorders of the heart and blood vessels. CVDs include ischemic heart diseases (IHD), cerebrovascular disease, peripheral vascular disease, rheumatic heart disease, valvular heart diseases, and congenital heart diseases [1]. CVDs are considered a major cause of mortality worldwide [2,3,4]. For instance, CVD contributed to 17.9 million deaths in 2019, with heart diseases and stroke contributing to 85% of that. In addition, 38% of the premature deaths attributed to non-communicable diseases are caused by CVD across the world [2]. Furthermore, disability-adjusted life years due to IHD and stroke have risen to 182 million and 143 million, respectively [5]. The above trend clearly demonstrates an overall increase in the burden of IHD worldwide. 

The burden of CVD is highest among low- and middle-income countries (LMICs) compared to high-income countries (HICs) [1]. Over 75% of CVD deaths occur in LMICs, with more premature deaths being attributed to CVD. For instance, in sub-Saharan Africa, over 50% of the deaths due to CVDs are among the age group of 30–69 years [6], which is nearly 10 years earlier than HICs, leading to losses of productivity and livelihood. The estimated economic loss due to CVD in LMICs was USD 3.7 trillion from 2011 to 2015. This represents 2% of the gross domestic product and contributed to nearly 50% of the non-communicable disease burden [7]. 

Cardiovascular diseases are closely associated with behavioral and socioeconomic factors. The majority of CVDs can be addressed in the early stages by modifying unhealthy behaviors. For instance, reducing tobacco consumption, modification of dietary risks, and increasing physical activity can be considered. The behavioral factors cause the development of intermediate-risk factors including hypertension, high plasma glucose level, hyperlipidemia, and high BMI. Also, poor governance, inefficient healthcare delivery methods, and insufficient funding to the health systems are major problems associated with poor healthcare delivery in LMICs, leading to a high prevalence of CVD [6]. Therefore, addressing the aforementioned disparities is important in order to reduce the global CVD burden. 

The prevention of CVD remains challenging for healthcare systems worldwide. According to the literature, cardiac rehabilitation (CR) stands as a major secondary prevention strategy of CVD [8]. Cardiac rehabilitation (CR) is a multidisciplinary intervention recommended for patients with chronic or post-acute cardiovascular events such as, following a myocardial infarction, coronary revascularization, heart transplant, acute or chronic angina, or heart failure. The overall goal of CR is focused on returning early to daily activities and reducing cardiovascular risk factors [9]. CR is a multi-interventional prevention method including risk factor modification, patient-tailored exercises, psychosocial counseling, and education [10]. 

Cardiac rehabilitation programs are mainly supervised by physicians such as cardiologists. Moreover, the team consists of a nurse, physiotherapist, occupational therapist, speech therapist, behavioral therapist, psychologist, dietician, and family members [11]. Physiotherapists help patients to develop individual exercise plans and dietitians are involved in helping patients maintain healthy eating habits. A social worker or psychologist helps patients to overcome stress and reduce any type of identified psychological condition, including smoking cessation programs [12]. 

Referral to a CR program is based on clinical diagnosis, and it involves three phases.

Phase I: Clinical phase

The first phase of the CR program starts during the hospitalization period, after a cardiovascular event or after an intervention like revascularization. The aim is to facilitate early ambulation and provide motivation for the rehabilitation process [11]. The CR team needs to pay attention to activities of daily living (ADLs), train the patient to reduce their stress during the CR process, and encourage the patient to remain active until the completion of the rehabilitation program [12]. 

Phase II: Outpatient Cardiac Rehab

The outpatient cardiac rehabilitation begins after identifying that the patient is stable via cardiology. Once three weeks have been completed in phase I, then phase II is started between the third and sixth week. Phase II is mainly based on the assessment of individual needs of patients, including electrocardiographic monitoring, counseling, and aggressive risk factor management. This phase promotes the healthy lifestyle of the patient and prepares them to return to their normal lives [11]. 

Phase III: Post-Cardiac Rehab

Phase III is conducted with minimal supervision or without any supervision by encouraging self-monitoring and periodic medical assessment of disease conditions. Phase III ensures lifelong commitment to maintaining healthy behaviors [9].

CR has been recognized as one of the best cost-effective treatments and secondary prevention methods that improve quality of life [13]. Evidence shows that CR increases the overall prognosis by reducing mortality, morbidity, and re-hospitalization by 20% [12]. According to the literature, the health-related quality of life (HRQoL) of patients after 12 months of engagement with CR has significantly improved. Randomized control trials (RCTs) by Anderson and his colleagues found that CR has reduced hospitalizations in post-myocardial infarction patients due to heart failure compared to their control groups [10]. CR can reduce the probability of mortality within the 5 years following a CVD or bypass surgery by 35% [14]. 

Further, CR programs help to increase the body’s functional capacity and decrease associated disabilities and weaknesses by providing patient-tailored physical exercises according to their medical diagnosis [10]. In addition to that, assuring psychological wellbeing is one of the major components of the CR program, helping to reduce stress and anxiety after a cardiac event, promoting faster recovery and reducing depression and other mental disorders [9]. Moreover, CR programs encourage and monitor adherence to medical management, thus improving patients’ compliance and reducing future mortality and morbidity due to cardiac events [15].

Despite CR being recognized as a gold standard treatment strategy, it remains under-utilized around the globe [12]. The availability of CR has been reported to be significantly low, at 60% in HICs, 28% in middle-income countries, and 8% in low-income countries, respectively. Further, low referral and participation rates and higher dropout rates have been reported globally [16]. 

The main objective of this scoping review is to identify the barriers to CR among patients diagnosed with CVD from a global perspective. Identifying these barriers will enable the respective authorities, healthcare providers, and stakeholders to focus on priority areas and implement sustainable measurements to enhance the availability of, referrals to, and participation in CR, thus reducing the global CVD-related mortality and morbidity.

## 2. Method

### 2.1. Study Design 

A scoping review of the literature was conducted to assess the barriers to cardiac rehabilitation among patients diagnosed with CVD according to the Joanna Briggs Institute (JBI) guidelines for scoping reviews [17]. The selection of studies following the initial database search was conducted using PRISMA-ScR (PRISMA for scoping reviews). See Figure 1.

### 2.2. Data Sources 

An extensive literature search was conducted using multiple databases, such as CINAHL, SCOPUS, EBSCOHost, and PubMed, from 2010 to 2023 to improve the results and reduce the risk of overlooking any eligible studies that could be used during our final appraisal [18]. The following search terms were utilized: barrier* AND (challenge OR limitation) AND (“cardiac rehab”) AND (“cardiovascular disease” OR CVD OR “heart disease”). The search was limited to patients diagnosed with CVD and studies published in English.

### 2.3. Study Selection

We included articles with the following criteria: (1) studies with patients diagnosed with CVD who were referred to and/or participated in and/or benefitted from CR programs; (2) studies that observed individual characteristics, hospital system-related factors, and socioeconomic factors as barriers to CR referral and/or participation and/or drop out among the study population; and (3) studies that compared the barriers in a global context, focusing on HICs and LMICs. Also, we included both qualitative (cohort, case–control, cross-sectional, and systematic reviews) and quantitative studies in this review. We excluded articles published before 2010 and those with small sample sizes (n ≤ 50).

### 2.4. Data Synthesis and Analysis

Data from selected studies were extracted according to the standardized approach and were sorted into tables. The study characteristics were recorded according to the study design, author, year, country, study subjects, barriers, and parameters (Table 1). The results were analyzed using thematic analysis. Following data extraction, the review adopted Braun and Clarke’s approach to thematic analysis to evaluate and summarize the data [19]. The results were categorized into three main themes, including healthcare system-related factors, socio-economic factors, and individual characteristics (Table 2). Extracted data under three main themes were categorized into subthemes for further interpretation. 

### 2.5. Ethical Consideration

As this study used secondary data from the literature, there were no ethical concerns.

## 3. Results 

### 3.1. Identification and Selection of Literature

The main three databases were divided among three researchers who searched the electronic databases separately using key terms, and a total of 2098 studies were found. Then, 1012 duplicates were removed. After that, articles were filtered according to titles and abstracts, and 803 were excluded. The final 14 articles were selected after excluding 269 articles by reading the full text. Fourteen studies were included in the scoping review. The characteristics of the included studies are summarized in Table 1. One study was a systematic review [20], four studies were prospective cohort studies [21,22,23,24], three studies were retrospective cohort studies [25,26,27], four studies were cross-sectional studies [28,29,30,31], and one study was a qualitative study [30]. Two globally conducted studies were analyzed in the scoping review. Other studies were from the United States, Canada, Iran, Netherland, Sweden, and Spain. Most of the participants were either patients diagnosed with CVD or CR providers, including cardiologists, physicians, and CR managers. Findings of included studies are illustrated in Table 2.

**Table 1 ijerph-21-00339-t001:** Characteristics of included studies.

Study Title	Author, Year, Country	Design	Study Subjects (n)	Barriers/Parameters
**Predictors of Early and Late Enrollment in Cardiac Rehabilitation, Among Those Referred, After Acute Myocardial Infarction**	Parashar et al., 2012, USA [24]	Prospective cohort study	1568	Age and gender Other chronic conditions, including smoking Patient’s education level Cost of care
**Smoking and Cardiac Rehabilitation Participation: Associations with Referral, Attendance, and Adherence**	Galeema et al., 2015, USA [20]	Systematic review	56 peer-reviewed articles	Other chronic conditions
**Factors Associated with Utilization of** **Cardiac Rehabilitation Among** **Patients With Ischemic Heart Disease** **in the Veterans Health Administration**	Schopfer et al., 2016, USA [32]	Qualitative study	56 patients, providers, and CR program managers	Lack of provider knowledge of the benefits and guidelines Inter-provider communication Cost of care Travel/distance Lack of patient desire
**Association of Mental Health Conditions with Participation in Cardiac Rehabilitation**	Krishnamurthi et al., 2019, USA [23]	Prospective cohort study	86,537 patients	Other chronic conditions
**Barriers for the Referral to Outpatient Cardiac Rehabilitation: A Predictive Model Including Actual and Perceived Risk Factors and Perceived Control**	Soroush et al., 2018, Iran [30]	Cross-sectional study	312 CABG patients	Age and gender Employment status Accessibility
**Factors associated with non-attendance at exercise-based cardiac rehabilitation**	Borg et al., 2019. Sweden [25]	Retrospective cohort study	31,297	Other chronic conditions Employment status Accessibility
**Association of Neighborhood Socioeconomic Context with Participation in Cardiac Rehabilitation**	Bachmann et al., 2017. USA [21]	Prospective cohort study	4096	Income Educational status Age and gender Other chronic conditions
**Effect of cardiac rehabilitation referral strategies on utilization rates**	Grace et al., 2011, Canada [22]	Prospective cohort study	1809	Referral strategies
**Cardiac Rehabilitation Availability and Density around the Globe**	Turk-Adawi et al., 2019, Global [31]	Cross-sectional study	98 countries	CR availability Referral strategy Mode of delivery
**Cardiac rehabilitation delivery in low-/middle-income countries**	Pesah et al., 2019, Global [29]	Cross-sectional study	55 countries	Availability Core components of the program Cost of care
**Physician-Related Factors Affecting Cardiac Rehabilitation Referral**	Moradi et al., 2017, Iran [28]	Cross-sectional study	122 Cardiologists	Physician’s knowledge about CR
**Referral and participation in cardiac rehabilitation of patients following acute coronary syndrome; lessons learned**	Rodrigo et al., 2011, Netherland [27]	Retrospective cohort study	469	Accessibility Other chronic conditions
**Predictors of Enrollment in Cardiac Rehabilitation Programs in Spain**	Chomsa et al., 2015, Spain [26]	Retrospective cohort study	756	Age and gender Other chronic conditions Accessibility

**Table 2 ijerph-21-00339-t002:** Results of included studies.

Author/Year	Themes	Results	Interpretation of Significant Findings
**Healthcare System-Related Factors**			
**Turk-Adawi et al., 2019** [31] **Pesah et al., 2019** [29]	Availability of CR programs	CR was available in 111/203 countries. Availability by region shows significant differences (*p* < 0.001) 5753 programs globally (χ2 =37.3, *p* < 0.001)	CR is available in 54.7% of countries worldwide 80.7% of countries in Europe to 17.0% in Africa Could serve 1,655,083 patients/year, despite an estimated 20,279,651 IHD cases globally/year CR is only available in 16.7% of LICs, 47.1% of MICs, and 86.2% of HICs There was one CR spot for every 66 IHD patients in LMICs (vs. 3.4 in HICs)
**Grace et al., 2011** [22] **Turk-Adawi et al., 2019** [31]	Referral strategies	(OR, 3.27; CI, 1.52–7.04) (OR, 3.35; CI, 1.54–7.29) (OR, 8.41; CI, 3.57–19.85) (OR, 1.36; CI, 1.35–1.38)	Automatic referral strategy resulted in 70.2% referral rate and 60% enrollment in CR Liaison referral strategy resulted in 59% referral rate and 50% enrollment Combined use of automatic and liaison strategies resulted in 85.8% referral rate and 73.5% enrollment Traditional referral strategy resulted in a 32.2% referral rate and 29% enrollment Systematic referral strategies resulted in 36% higher referral rates compared to traditional referral strategies.
**Schopfer et al., 2016** [32] **Moradi et al., 2017** [28]	Providers’/physicians’ knowledge	73%—CR providers 79.5%—cardiologists	73% of CR providers perceived a lack of knowledge regarding the benefits and guidelines, which causes fewer referral rates to CR 79.5% of cardiologists perceived low general knowledge about CR programs as the standard of care which had an impact on referral to CR
**Schopfer et al., 2016** [32]	Inter-provider communication	18%—CR providers 17%—CR managers	18% of CR managers and 17% of providers perceived poor communication between clinicians regarding patients’ eligibility for CR, which resulted in fewer referrals
**Turk-Adawi et al., 2019** [31]	Mode/setting of delivery	(OR = 1.05, 95%CI = 1.04–1.06)	CR programs offered individualized consultation with physicians, which reported high participation rates, with residential programs reporting higher patient compliance
**Socioeconomic factors**			
**Parashar et al., 2012** [24] **Bachmann et al., 2017** [21] **Soroush et al., 2018** [30]	Level of education	1st month (OR, 1.38; 95% CI, 1.04–1.84) After 6 month (OR, 1.81; 95% CI, 1.42–2.30 Completed high school—(OR, 1.20; 95% CI, 0.92–1.58) Completed college—(OR, 1.61, 95% CI, 1.06–2.44) Illiterate—7% Less than diploma—9% Academic—16%	People who have at least high school education had 38% higher participation at 1st month and 81% after 6 months of AMI People who have completed college had 61% higher participation in CR compared to people who completed high school Higher referral rate (16%) for CR among people who completed academic education
**Parashar et al., 2012** [24] **Pesah et al., 2019** [29] **Schopfer et al., 2016** [32]	Cost of care	Uninsured (first month) (OR, 0.39; 95% CI, 0.21–0.71) After 6 months insured vs. uninsured *p* < 0.001 Economic burden (first vs. sixth month) (OR, 1.48; 95% CI, 0.97–2.26). vs. (OR, 0.56; 95% CI, 0.38–0.81) LMICs vs. HICs Out-of-pocket (n = 212, 65.0%) vs. (n = 184, 24.9%) 27% of participants perceived cost of care as a barrier	Uninsured patients were 40% less likely to participate in the first month and no significance was found in insured vs. uninsured patients at 6 months Patients with economic burden showed 48% higher participation in the first month, but 44% less participation at 6 months High out-of-pocket expenditure was significantly associated with less participation and high dropout rates in LMICs compared to HICs 27% perceived higher cost of CR program reduce participation
Soroush et al., 2018 [30] **Borg et al., 2019** [25] **Chomsa et al., 2015,** [26] Bachmann et al., 2017 [21]	Employment status/income	Employed—23% Personal job—6.6% Retired—12% Unemployed—3.7% Employed vs. retired (OR, 0.86; CI, 0.80–0.93) Self-employed (OR = 1.56; 95% CI: 0.62–3.92) Retired (OR = 1.33; 95% CI: 0.62–2.77). <USD 15,000 vs. >USD 25,000 (OR 1.68, 95% CI, 1.17–2.42)	Unemployed, retired, or self-employed patients were less likely to be referred to CR than employed patients. Retired patients are 14% less likely to participate in CR Both self-employed (56%) and retired patients (33%) have higher enrollment Those with household incomes >USD 25,000/y had 68% higher participation
**Schopfer et al., 2016** [32] Soroush et al., 2018 [30] **Rodrigo et al., 2011** [27] Borg et al., 2019 [25] **Chomsa et al., 2015,** [26] Bachmann et al., 2017 [21] **Fraser et al.,** [33]	Accessibility to CR facilities	68% responded to travel issues as a barrier Distance to CR center (*p* < 0.042) <5 km vs. > 20 km referral (OR 4.0; CI1.26–13.0) participation (OR 0.2, CI 0.07–0.79, (OR 1.75 [95% CI: 1.64–1.86] (OR = 2.87; 95% CI: 1.29–6.41) (OR 0.71, 95% CI, 0.59–0.84)	The most perceived barriers to CR participation were long distance and transportation issues Larger distance was significantly associated with fewer referrals Larger distance (>20 km) to CR centers led to a 4 times higher referral rate, but their participation in CR was significantly low Distance >16 km increased non-attendance by 75% Distance to CR unit >50 km led to a likelihood of CR non-enrollment threefold higher An increase in distance to CR centers from 3.8 km to 25 km reduced the attendance by 29%
**Individual characteristics**			
**Parashar et al., 2012** [24] **Chomsa et al., 2015,** [26]	Age	OR, 0.85 for each 10-year increment; 95% CI, 0.74–0.97 (OR = 1.05; 95% CI: 1.02–1.09).	Older patients were 15% less likely to participate in CR Age was associated with no enrollment, and the chance of not enrolling increased by 5% for every year of age
**Parashar et al., 2012** [24]: Borg et al., 2019 [25] **Chomsa et al., 2015,** [26]	Gender	(OR, 0.61; 95% CI, 0.44, 0.86) (Female vs. male) OR, 0.85; 95% CI, 0.80, 0.90 Female vs. male (20.8% vs. 35.9%) Women with MI (OR, 6.35; CI, 2.53–11.81)	Women were 40% less likely to participate in CR Men were 15% less likely to participate Referrals were fewer among women Women with MI had 35% higher non-participation
Parashar et al., 2012 [24] Borg et al., 2019 [25] **Krishnamurthi et al., 2019** [23] **Chomsa et al., 2015,** [26] **Galeema et al., 2015** [20] Bachmann et al., 2017 [21]	Comorbidities and individual behaviors	Hypertension (OR, 0.58; 95% CI, 0.43–0.78), PAD (OR, 0.43; 95% CI, 0.22–0.85), and previous PCI (OR, 0.55; 95% CI, 0.36–0.83) Smokers (OR, 0.59; 95% CI, 0.44–0.80) Diabetes (OR, 1.20; 95% CI, 1.13–1.28) Hypertension (OR, 0.94; 95% CI, 0.89–0.98) Smoking (OR, 1.63; 95% CI, 1.54–1.74) (OR, 1.57; 95% CI, 1.43–1.74) (OR: 6.35; 95% CI: 2.53–11.81) (OR, 0.59; 95% CI, 0.44–0.80 (OR 0.65, 95% CI, 0.49–0.85,)	Patients with a greater number of comorbidities were less likely to participate in CR Non-attendance at CR was higher for individuals with a higher burden of comorbidities and for smokers Patients with both PTSD and depression had 57% greater odds of participating in CR than those without depression or PTSD Women with previous MI were less likely to participate Smokers were less likely to participate Smokers were 35% less likely to participate in CR programs

### 3.2. Healthcare System Related Factors

#### 3.2.1. Availability of Cardiac Rehabilitation Programs

Three studies examined the availability of cardiac rehabilitation worldwide [29,31,32]. Turk-Adwai and colleagues conducted a cross-sectional study across the globe, including 98 countries, to determine the availability, capacity, and density of the cardiac rehabilitation programs. The study identified that 54.7% of the countries worldwide had cardiac rehabilitation programs. Furthermore, disparities in availability were identified across the regions. For instance, cardiac rehabilitation programs were available in 80.7% of European countries compared to 17.0% of African countries (*p* < 0.001). Moreover, 5735 CR programs were identified, and they could facilitate only 1,655,083 patients out of 20,279,651 annual IHD patients worldwide [31]. Another cross-sectional study was conducted by Pesah and colleagues [29] in 55 countries to identify the barriers to the delivery of cardiac rehabilitation in LMICs. According to their study, cardiac rehabilitation programs are available in 16.7% of LICs, 47.1% of MICs, and 86.2% of HICs (χ2 =37.3, *p* < 0.001). Moreover, in LMICs, only one spot in a cardiac rehabilitation program was available for 66 IHD patients.

#### 3.2.2. Referral Strategies 

Two studies focused on the impact of referral strategies on the utilization rates of cardiac rehabilitation [22,31]. A prospective cohort study conducted by Grace and colleagues with 1809 participants has concluded that automated referral strategies lead to a 70% referral rate and 60% enrollment in cardiac rehabilitation (OR, 3.27; CI, 1.52–7.04). The liaison referral strategy resulted in a 59% referral rate and 50.6% enrollment in cardiac rehabilitation (OR, 3.35; CL, 1.54–7.29). Moreover, the combined use of automated and liaison referral strategies has shown 85.8% referral rates and 73.5% enrollment (OR, 8.41; CI, 3.57–19.85) compared to traditional referral strategies, which only resulted in a 32.2% referral rate and 29% enrollment [22]. Similar results were identified by a cross-sectional study conducted by Turk-Adwai and colleagues regarding the availability and density of cardiac rehabilitation programs worldwide. They also identified higher referral rates where systematic referral strategies were used compared to programs where traditional referral strategies were used [31]. 

#### 3.2.3. Providers’ Knowledge of CR

Two studies showed the effect of cardiac rehabilitation providers’ knowledge on referral rates for cardiac rehabilitation [28,32]. A qualitative study conducted by Schopfer and colleagues found that 73% of cardiac rehabilitation providers were unfamiliar with the indications and referral strategies of cardiac rehabilitation, leading to lower referral rates [32]. Moradi and colleagues [28] conducted a cross-sectional study among 122 cardiologists to observe the physician-related factors affecting cardiac rehabilitation. A total of 79.5% of cardiologists reported that a physician’s lack of knowledge of benefits, program attributes, and referral strategies acts as a barrier to cardiac rehabilitation. 

#### 3.2.4. Inter-Provider Communication

Inter-provider communication methods and delivery modes or settings were found to have an impact on referral rates and participation rates of cardiac rehabilitation [31,32]. A qualitative study conducted by Schopfer and colleagues to identify the patient- and provider-level barriers and facilitators to cardiac rehabilitation showed that poor inter-provider communication results in lower referral rates. In total, 18% of CR providers and 17% of CR managers have perceived poor communication among clinicians and providers as a barrier to cardiac rehabilitation [32]. Turk-Adwai and colleagues’ cross-sectional study, which was conducted to observe the availability and density of the cardiac rehabilitation programs, found that the mode/setting of delivery affects participation rates (OR = 1.05, 95% CI = 1.04–1.06) [31]. 

### 3.3. Socio-Spatial Factors

#### 3.3.1. Accessibility to Cardiac Rehabilitation Facilities

Six studies provided evidence on the association of accessibility to CR facilities and patients’ enrollment and participation [21,25,26,27,30,32]. Borg and his colleagues conducted a retrospective cohort study to identify the factors related to non-attendance in center-based CR programs using the data of 31,297 post-AMI patients, which covered a six-year period [25]. They identified the distance to a CR facility as the strongest predictor for non-attendance (OR 1.75 [95% CI: 1.64–1.86]), where patients who lived >16 km away from facilities had 25% higher non-attendance compared to their counterparts. Also, the prospective cohort study conducted by Bachman and his colleagues found that an increase in the distance to the CR center significantly reduced the CR participation by 29% (OR 0.71, 95% CI, 0.59–0.84, *p* < 0.001) [21]. Chamosa and colleagues [26] also conducted a retrospective cohort study on 756 patients with myocardial infarction from 2009 to 2012. According to the study, they found that patients living >50 km away from CR unit have threefold higher chances of non-enrollment. Similarly, Rodrigo et al. [27] conducted a study on 469 patients who were hospitalized for acute coronary syndrome in 2017 and evaluated the predictors for their CR referral and participation. They found that patients living a >20 km distance from the facility had higher referral rates (OR 4.0, CI 1.26–13.0), but less participation (OR 0.2, CI 0.07–0.79) compared to patients who lived a <5 km distance from the CR center [27]. Moreover, a cross-sectional study conducted on 312 CABG patients by Soroush and colleagues to assess the predictors for CR referral found that long distances to CR facilities significantly reduce the referral rates (*p* < 0.042) among CABG patients [30]. A qualitative study was conducted by Schopfer and his colleagues with a total of 56 participants, including patients, CR providers, and program managers, to assess the barriers to CR utilization. According to their results, the most perceived barrier involved problems with transportation and distance to CR centers (68%), which limited the utilization of CR [32].

#### 3.3.2. Employment Status/Income 

Four studies found an association between employment status and enrollment and attendance to CR programs [21,25,26,30]. The study conducted by Borg and his colleagues [25] found that retired patients, compared to employed patients, were 14% less likely to attend CR programs (OR 0.86; CI,0.80–0.93) (Borg et al., 2019). According to the prospective cohort study by Bachmann et al., a household income >USD 25,000 per year leads to 64% higher participation (OR 1.68, 95% CI, 1.17–2.42, *p* < 0.01) compared to a household with <USD 15,000 per year [21]. Another study {30] found that there was a higher rate of CR referral among patients who were employed (24%) compared to retired (12%) and unemployed (4%) patients. In contrast, one study [26] found no significant differences between the rates of enrollment among self-employed (OR = 1.56; 95% CI: 0.62–3.92) and retired workers (OR = 1.33; 95% CI: 0.62–2.77).

#### 3.3.3. Level of Education

Three studies found an association between the level of education and CR participation [21,24,30]. According to Bachmann et al. [21], people who had completed higher/tertiary education had higher participation rates (OR 1.61, 95% CI, 1.06–2.44, *p* < 0.05) compared to those who had completed a high school education. Another study [30] identified that illiterate patients were less likely to refer to CR programs compared to patients who had higher educational levels (7% and 16%, respectively). Similarly, a prospective cohort study [24] on 1568 patients with AMI looked at the CR participation within 1 month and 6 months post-MI and found that patients who had completed at least a high school education showed higher participation in the first month post-MI (OR, 1.38; 95% CI, 1.04–1.84) and 6 months after the MI event (OR, 1.81; 95% CI, 1.42–2.30).

#### 3.3.4. Affordability and Access to Care

Three studies found an association between the cost of care and CR participation [24,29,32]. Pesha and his colleagues conducted a cross-sectional study collecting data from all available CR programs globally and analyzing the relation between cost of care and CR participation and dropout [29]. According to their study, in LMICs, 65% of CR costs consisted of out-of-pocket expenditures, and this was 24% in HICs. Also, the prospective cohort study of Parashar et al. [24] found that uninsured patients (OR, 0.39; 95% CI, 0.21–0.71) were 60% less likely to participate in CR in the first month of the post-MI period, but the insurance status of the patient (both uninsured and insured, *p* < 0.001) did not predict CR participation at 6 months [24]. In addition, patients with economic burdens showed lower participation in CR programs after 6 months (OR, 0.56; 95% CI, 0.38–0.81) compared to the first month (OR, 1.48; 95% CI, 0.97–2.26).

### 3.4. Individual Characteristics 

#### 3.4.1. Comorbidities and Individual Behaviors 

Six studies found associations between comorbidities and lower participation in CR programs [20,21,23,24,25,26] Borg and colleagues [25] identified smoking (OR, 1.63; 95% CI, 1.54–1.74), history of stroke (OR,1.37; CI, 1.21–1.550 percutaneous coronary intervention (PCI) (OR, 1.28; CI, 1.16–1.42), coronary artery bypass graft (OR,1.31; CI,1.16–1.42, AMI (OR,1.19; CI, 1.08–1.31), and diabetes (OR,1.20; 95% CI, 1.13–1.28) as predictors for non-attendance in CR. The prospective cohort study [24] found that lower participation rates were reported after one month by patients with hypertension (OR, 0.58; CI, 0.43–0.78) and peripheral arterial disease (OR, 0.43; CI.0.22–0.850). Smokers were also less likely to participate after six months (OR, 0.59; CI, 0.44–0.80) [21]. Bachmann et al. [21] also found similar results, with smoking patients being less likely to participate in CR (OR, 0.65; CI, 0.49–0.85). Furthermore, a systematic review [20] evaluated 56 peer-reviewed articles to determine the association between smoking and referral, attendance, and adherence to CR. The results found that smokers were more likely to obtain a referral to CR, but showed less participation and non-adherence to CR [20]. In addition, Krishnamurthy et al.’s prospective cohort study, which looked at the association between mental health conditions and participation in CR, the study found that patients with post-traumatic stress disorder (PTSD) and depression were more likely to participate in CR (OR, 1.57; CI, 1.43–1.74) [23].

#### 3.4.2. Generic Factors

Two studies found an association between older age and lower participation rates in CR [24,26]. Chamosa and associates found that the older age group had higher non-enrollment, and with every year of age, there was a 5% chance of rising non-enrollment (OR, 1.05; 95% CI: 1.02–1.09) [26]. The cohort study by Parashar et al. (2012) found that older patients were 15% less likely to participate in CR (OR, 0.85 for each 10-year increment; CI, 0.74–0.97) [24].

These studies also showed a relationship between the female gender and lower participation rates [24,26]. Chamosa’s cohort study [26] found that women with previous MI were less likely to participate (OR: 6.35; 95% CI: 2.53–11.81) and less likely to refer to CR, whereas referrals for men were recorded to be 35.9%, and for women, 20.8%. Conversely, one study found male gender as a predictor for non-attendance in CR [25].

## 4. Discussion

The results of the current scoping review suggest that healthcare system-related factors, socio-economic factors, and individual characteristics have a significant impact on referral, enrollment, and participation in cardiac rehabilitation programs. 

Healthcare system-related factors, including poor availability of CR programs, lack of proper referral strategies, inadequate knowledge of CR providers, inter-provider communication issues, and lack of alternative methods of CR delivery were identified as barriers to CR.

Despite CR being widely recognized as an important intervention in CVD management, this review identified disparities in the availability of CR programs across the globe, particularly in under-privileged communities and resource-poor settings. The need to establish more rehabilitation services for both communicable and non-communicable diseases has been identified worldwide. For example, a study conducted to estimate the need for rehabilitation services identified that 2.41 billion people could benefit from rehabilitation programs if access were provided [34]. 

Traditional referral strategies resulted in lower referral rates and enrollment rates compared to systematic referral strategies. Similarly, a study conducted by Gravely-Witte and colleagues identified that automatic and liaison referral strategies could considerably increase the referral and participation rates [35]. Furthermore, proper communication among CR providers, including in-patient cardiologists, follow-up specialty clinics, and primary care and CR services, can lead to higher referral rates to rehabilitation centers. In addition, CR programs delivered via alternative modes, such as home-based rehabilitation and tele-rehabilitation, result in higher patient compliance in rehabilitation. According to Ozemek and colleagues, CR services can be delivered remotely using alternative methods to increase CR participation [36].

This scoping review found that socioeconomic factors such as lack of accessibility to CR facilities, a low level of education, unemployment or poor income, and high cost of care were identified as barriers to referrals to and participation in CR. This is consistent with other studies which have emphasized the impact of geographic location and resource allocation on the completion of CR programs, such as Fraser et al. [33].

A greater distance to CR facilities has been identified as a major barrier to CR referral and participation. Several contributing factors, like the unavailability of proper transportation methods, cost of transportation, and time constraints, may add to the reluctance to travel long distances [37]. Therefore, proper access to healthcare has a great impact on health outcomes, especially for non-communicable chronic diseases like CVD, where follow-up and prevention methods such as CR are crucial. A systematic review conducted by Starbird and colleagues found a significant association between establishing transport interventions and an increase in chronic care management services [38]. Also, encouraging home-based CR programs can resolve the problems with access to care, thus increasing participation and adherence [39]. It should be noted here that individual-level barriers to rehabilitation and healthy aging are principally influenced by the socio-economic status of the country and the level of investment in healthcare, which collectively reduce access to and quality of services overall, particularly in resource-poor settings [40].

Patients’ levels of education play a significant role in CR participation and enrollment. The individual level of education plays a crucial role in the literacy of the person. Acquired knowledge of individuals is used to understand and make positive decisions related to health outcomes, to acquire health-related information, and for health promotion and providing care [41]. For example, a cohort study conducted by Demmler and his colleagues found that children who have successful education attainments have fewer health-risk behaviors during the adolescent period, proving the positive influence of education on health [42].

Other factors that affect CR participation and referral are poor income status, unemployment, and the high cost of rehabilitation sessions. Health outcomes of individuals highly depend on income status, as income is the major indicator for acquiring all resources in daily life, like food, medicine, housing, healthcare services, and education [43]. Again, the socio-economic status of the country directly impacts this factor, not merely the individual income.

Also, comorbidities such as a history of stroke, PCI, AMI, CABG, diabetes, hypertension, low physical function, and smoking, as well as female gender and older age, were identified as the individual characteristics associated with less enrollment and participation in CR. In line with the literature, a lack of desire and lack of willingness of the patient to participate in CR also restrict engagement. Following the revascularization procedure, some patients think that they will be cured, with some having a lack of trust in the benefit of CR [32]. As comorbidities reduces one’s functional capacity, tailored CR programs can be planned and implemented to improve participation rates [44]. Moreover, smoking stands as a strong barrier to enrollment and participation, prompting the need for a specific approach for these groups of patients [20]. In contrast with the literature, a study by Krshnamurthi et al. [23] showed that patients with comorbid depression and PTSD have higher participation rates. This may be due to CR providing more mental health support to these patients, which would subsequently enhance the overall engagement with the rehabilitation.

The female gender showed disparities in referral, enrollment, and participation. Women’s health-seeking behaviors for their symptoms are postponed compared to men, and their exercise tolerance is lower than men’s, especially following a cardiac event [45]. Gender-specific cardiac health needs for women are recognized as an effective intervention for women. In line with the literature, gender-tailored CR programs showed improved attendance by 31% [44]. Gender disparity in referral enrollment and participation in CR can be addressed by planning gender-tailored programs according to the unique needs of females. Different findings were presented by Borg et al. [25], who found that female participation was higher. However, their study was limited to exercise-based CR programs and was based in Sweden, a non-male-dominant country.

Independent of gender, older patients’ enrollment and participation were reported to be lower. When compared with younger patients, older patients generally have lower incomes, and they may not perceive CR as a beneficial approach. However, according to recent literature, younger patients and male patients show lower participation rates [21,25], whereas CR achievements should be planned for both sexes and all age groups by addressing other influencing factors.

## 5. Limitations

This review has a few limitations. Even though the database search was conducted at a global level, the studies were limited to a few countries, including the USA, Canada, the Netherlands, Sweden, Spain, and Iran. Therefore, evidence from LMICs was limited, and the outcomes of the scoping review were mostly based on HICs. However, some of the studies in HICs managed to address social disadvantages and inequity in access to health services.

Also, due to time constraints, the scoping review method was used to conduct the analysis. However, if a systematic review and meta-analysis were conducted, a more robust global level evidence could have been synthesized, and a better understanding of the barriers to CR could have been obtained.

## 6. Recommendations and Future Directions

More rigorous, global-level studies should be conducted to identify barriers to CR in depth, particularly in LMICs. Governments should allocate more funding to establishing home-based and community-based cardiac rehabilitation services to enhance participation and adherence. Also, governments should consider increasing publicly-funded programs to reduce out-of-pocket expenditures in CR. Further, introducing advanced technologies to increase the utilization of automated referral strategies and to promote user-friendly telehealth services is recommended. Additionally, continuous regular training should be provided for healthcare professionals to increase the efficacy of CR delivery, to review patients’ understanding of CR, and to incorporate education sessions to CR to increase patient compliance.

Nevertheless, for more sustainable tightening of the gap in access to health services between low- and high-income groups, issues of social injustice and inequity must be addressed and tackled. Governments and decision makers should aim at implementing comprehensive strategies to address the root causes of disparities and promote equity in healthcare access and outcomes. Improving access to health, particularly for vulnerable groups, would require tailored interventions and programs that aim at addressing social determinants of health. A more cost-effective approach could also include programs that focus on promoting health in these groups and investing in community empowerment and engagement [46]. More focused health literacy programs should target those who are in need of health literacy, and healthcare providers should invest more time and efforts into culturally and linguistically diverse groups [46,47,48] to enhance their engagement and adherence to post-treatment plans.

## 7. Conclusions

The burden of CVD is rising worldwide. Despite CR being considered as the most cost-effective treatment strategy for CVD, it is grossly underutilized. Various factors related to the healthcare system, socioeconomic status, and individual characteristics were identified as key barriers to cardiac rehabilitation. These findings should be the focus of healthcare planning and healthcare policies in an attempt to improve the outcomes of cardiovascular rehabilitation programs. Further studies are required in order to identify the barriers in depth, especially in areas with high prevalence of CVD, like LMICs.

## Figures and Tables

**Figure 1 ijerph-21-00339-f001:**
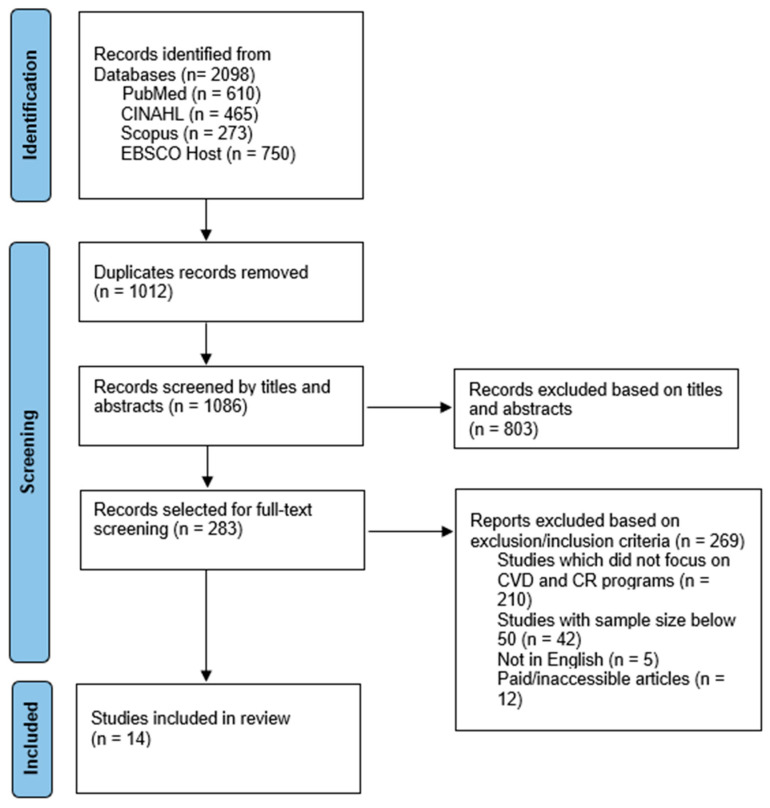
PRISMA flow diagram of study selection.

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
