# Peer review of "Barriers to Cardiac Rehabilitation among Patients Diagnosed with Cardiovascular Diseases—A Scoping Review"

_ijerph, 2024, doi:10.3390/ijerph21030339_

Round 1

Reviewer 1 Report

Comments and Suggestions for Authors

Highly relevant topic, the aim was clear and specific, relevant search terms, and methodology were clearly described, and findings were well presented.

Some issues with the organization of the text.

Lines 13 and 14: Mentioned Three databases but listed four (four were also listed in the flow chart)

The headings and numbers need major revision. 

Results should read Results and Discussion as the results were discussed with the data.

Line 180:  Mentioned 13 studies (instead of 14 stated in the abstract and flowchart)

Line 182-183: The Gannt Chart in that section does not seem to serve any purpose there. If at all, maybe in the Methods section as it describes the process of the review.

The numbering of Results and other sections should be revised.

For example, line 184 is 3.2. the subsection under it should be 3.2.1.

Line 184: 3.2. Healthcare System-Related Factors 

Line 185: 3.2.1. (not 3.3) Availability of Cardiac Rehabilitation Programs 

Socio-Spatial Factors should be 3.3

Individual Characteristics should be 3.4

The limitation section should be 4

The recommendation should be 5 

The conclusion should be 6

Comments on the Quality of English Language

Good and logical flow of thought. Require minor revision. Some sentences require minor revisions to make them more concise.

Author Response

Authors responses to R1

1-      Thank you, much appreciated positive remarks

2-      Thanks for picking this up and yes we have used 4 databases not 3. We have now corrected this and made it consistent throughout the paper

3-      Thank you for this and we have now revised this issue with headings. We did not originally have numbering and this was caused by the journal formatting but all is now adjusted.

4-      We have looked at this and although the results section somewhat lays out the findings, the subsequent discussion section tells the reader what it all means. We hope the reviewer will accept our apology for leaving the layout as is as the “results” section does not deeply discuss the above point.  We also tried to remove any “discussion” of findings from results section.

5-      We have now fixed the typographic issue, thank you for the pick-up.

6-      We agree with the reviewer, the Gannt chart is unnecessary and has now been omitted from the paper.

7-      We did not have any numbering in our manuscript and this was created by the journal formatting system and hence the out-of-order sequence. We have now adjusted the numbering. Thank you for this observation

8-      Thank you. We have further revised the manuscript and adjusted and tidied up any language issue

Reviewer 2 Report

Comments and Suggestions for Authors

Comments

The study has social and scientific value since it addresses a disease that is considered relevant at the level of epidemiological transition in medium to highly developed countries. Some countries are still in transition between infectious diseases and chronic diseases, having to consider strategies for both scenarios.

However, since it is a study that tries to approximate, I consider that some statements are not supported by the findings. Many of them will depend on the level of transition that the countries have and the level of GDP per capita that each country dedicates to health. Therefore, individual results cannot be extrapolated to global results because this would be committing an atomistic fallacy. This fallacy is committed when inferring about group variability based on individual-level data. In other words, it is assumed that what is true at the individual level is also true at the group level, without considering contextual differences.

Given the differences found and the relationships with the different contexts of the social determinants, it is complex to give recommendations based only on these results since for one it would be appropriate and for another it would not be. For example, not all populations have access to the Internet and at a certain age not all have access to the Internet or social networks. This means that some initiatives such as telehealth that seek to provide recommendations for the control of chronic diseases to elderly patients fail to reach a part of this population because they do not have access to this medium, which is why they are excluded. The same would happen with migrants, low-income people and others similar who are out of reach.

SUMMARY: It would be important to try to narrow down some answers to what the information is but try not to extrapolate these results due to the characteristics of the study design.

Author Response

1-      Great point and thanks for the positive remarks re importance of our study. When we started the work on this study, the idea was to address social inequity and its influence on disease and recovery etc. we feel that this has been fulfilled given our research question. In order to address social injustice, much more needs to be done including implementation of comprehensive strategies aimed at addressing the root causes of disparities and promoting equity in healthcare access and health outcomes. Some of this has now been added to recommendations in this review – see highlighted new section in recommendations

2-      We agree with the reviewer that health expenditure and investment in health are detrimental to access and that low SES/income countries maybe mostly impacted by this, however, in this review we refrained from analyzing that aspect and we hope to address the health economic issues impacting rehabilitation in a future work. We like to also add here that our findings in this review are solely based on the findings of individual studies rather than our own full interpretation, however, taking the reviewer’s important point, we have added a line in our discussion pointing out that fact “barriers to rehabilitation may largely be influenced by the economic status of the country and investment in health which significantly impact access and quality of services.” This is now added and highlighted in the paper.

3-      Interesting and rather valid point. We had that thought while writing the recommendation and we felt the ambiguity of the situation too. However, the issue of access to telehealth or other platforms can be restricted for some, example, senior populations, which is also the point of this analysis in this section. If we say that telehealth or accessing online services are a challenge for senior patients (which most of cardiac patients are) then that can pose an extra challenge on access to rehabilitation when other barriers are present mainly due to socio-economic issues.

4-      We have attempted to revise accordingly. The reviewer is correct, findings should not be over-interpreted so what is reported is exactly the summary of what has been reported by individual studies. These adjustments are reflected in the manuscript as highlights

Reviewer 3 Report

Comments and Suggestions for Authors

The proposed manuscript is an extremely interesting one, which draws attention to aspects associated with the adherence of patients with various cardiovascular pathologies to cardiovascular rehabilitation programmes. The manuscript needs to be improved in order to be considered for publication:

Abstract - line 15 - I think that the search should have been made by the word rehabilitation and not the abbreviated version

Introduction - references are not formatted according to journal recommendations.

Material and method - section 2.2 - use of abbreviations is not recommended. Please specify the number of studies identified by using X and not Y.  Please specify the main elements assessed in the studies at length.

Table 1 - detail what demographic factors, comorbidities

It would be useful to include an additional paragraph on potential solutions to the problems identified.

The number of references is far too small.

Author Response

1-      We thank the reviewer for the positive feedback. We will consider anything they may suggest and as fits

2-      This is a valid point and we consulted with a librarian on this and the recommendation was to use the Asterisk as a way of capturing articles that contain rehab… including “rehabilitation”, “rehabilitative therapy”, “rehabilitative interventions/programs/services” or “rehab” alone and possibly other terms. 

3-      Thank you, this has now been done throughout the manuscript to fit in Journal requirements

4-      Please refer to point no. 2 above for this. The abbreviated format with the use of “*” is the library attempt to capture more studies. We apologise to the reviewer but we did not quite understand the part of the comment “Please specify the number of studies identified by using X and not Y?” which section was this for?

5-      This has now been modified to fulfill the reviewer’s request

6-      A recommendation paragraph is naturally added. This was further developed to satisfy the reviewer’s comment. See additions in highlights

7-      This was primarily due to the available evidence on the specific topic. We have run the search multiple times and while few studies may have been missed, our search was quite rigorous and hope this will suffice for the purpose of this scoping review. However, we have now further enhanced our reference list to support our protocol and recommendations in addressing the disparity of health and the adoption of community-focused programs to address inequity of access and health in general.  

Round 2

Reviewer 3 Report

Comments and Suggestions for Authors

The manuscript has been improved and can be considered for publication. Discussion paragraph - attention to paragraph formatting and font used

I suggest finding another name for section 6.

I congratulate the authors for their work in identifying the limiting issues in the cardiac rehabilitation of patients with various cardiovascular pathologies.

Author Response

We thank the reviewer for the follow-up comments and appreciate their positive remarks on the improvement of our manuscript. We also agree with the review that formatting was required for the discussion section and we have now done that to fulfill their wish. Section 6 title has also been modified and is now called Recommendations and Future Directions.

Updated clean version of the manuscript is attached